# Multiple-Traits Selection in White Guinea Yam (*Dioscorea rotundata*) Genotypes

**DOI:** 10.3390/plants11213003

**Published:** 2022-11-07

**Authors:** Prince Emmanuel Norman, Paterne A. Agre, Robert Asiedu, Asrat Asfaw

**Affiliations:** 1Sierra Leone Agricultural Research Institute, Tower Hill, Freetown PMB 1313, Sierra Leone; 2International Institute of Tropical Agriculture, Ibadan PMB 5320, Nigeria; 3West Africa Centre for Crop Improvement, College of Basic and Applied Sciences, University of Ghana, Legon, Accra P.O. Box LG 30, Ghana

**Keywords:** *Dioscorea rotundata*, parental selection, agronomic traits, selection gain, mixed models

## Abstract

Choosing superior parents with complementary trait values for hybridization and selecting variants with desired product profiles to release as a new cultivar are important breeding activities to progress genetic improvement in crops. This study assessed the genetic potential of 36 parental lines of white Guinea yam (*Dioscorea rotundata*) genotypes using multi-trait index-based factor analysis and ideotype design (FAI-BLUP). The experiment utilized 36 white yam genotypes laid out in a 6 × 6 triple lattice design with three replications and phenotyped for 18 agronomic and food quality traits. Findings showed significant differences among genotypes for all assessed traits. Fifteen traits had desired genetic gains, whereas stem diameter (−1.34%), and two starch property traits ((holding strength (−26.31%) and final paste viscosity (−3.33%)) had undesired selection gain. The FAI-BLUP index provided total genetic gains of 148.91% for traits desired for increase and –29.26% for those desired for decrease. Genotypes TDr08-21-2, TDr9518544, TDr9501932, TDr8902665 and Pampars were identified as top best candidate for simultaneous improvement of the measured traits in white yam breeding. The findings indicate the effectiveness of the FAI-BLUP index in identifying and selecting genotypes.

## 1. Introduction

Yam (*Dioscorea* spp.) is a clonally propagated tuber crop belonging to the genus Dioscorea in the Dioscoreaceae family. It comprises >600 species, of which 11 are economically important and widely cultivated species [1,2]. The crop is produced on >8.5 million ha of land in about 61 countries distributed across tropical and subtropical Africa, Asia, Oceania, and Latin America [3]. White Guinea yam (*D. rotundata* Poir.) is the most cultivated and produced yam species in West Africa with high importance for production [4]. Despite its importance, several biotic and abiotic factors limit the full potential of the crop as food, feed, industrial application and source of income [5,6]. Biotic and abiotic stress factors contribute to low yields and poor market quality tubers of yams [1]. Moreover, biotic factors such as yam mosaic virus cause severe yield losses, and restrict international movement and exchange of germplasm [7]. These stress factors are a challenge in yam cultivation but could be resolved through breeding for better varieties.

Genetic improvement and deployment of elite yam genotypes are cyclic and incremental processes that demand discovery and selection of novel alleles and traits in germplasm, stacking elite allele and trait combinations in desired genetic background. Mixed models are often utilized for determination of predicted genotypic values for most important breeding objective traits and improvement of selection efficiency [8]. The restricted maximum likelihood (REML) and the best linear unbiased prediction (BLUP) have been utilized as efficient selection procedures for the estimation of variance components and prediction of genotypic values (random effect), respectively [9]. These techniques have been utilized in various crop breeding programs such as white Guinea yam [5], white oat [10], soybean [11], bean [12], maize [13], etc.

Genetic gain is key performance measure of success in plant breeding programs. Genetic gain refers to the difference in the mean values of the selection traits between the original generation and next generation which is formed from only the selected individuals [14]. Selection based on only a few traits is not the most appropriate strategy due to lack of assurance of genetic gains in other important traits [15]. Therefore, breeders often assemble various desirable traits in one new genotype that result in high performance [16]. This has led to the proposition of several selection indices targeted at selecting superior genotypes [17]. However, expression of the economic value of traits and their conversion into realistic economic weightings may limit breeders’ ability to identify and select the best genotypes when an inefficient selection index is utilized [18]. To overcome this limitation, Rocha et al. [19] suggested a multi-trait selection index based on factor analysis and genotype-ideotype distance (FAI-BLUP index). The FAI-BLUP index technique that involves the estimation of multi-trait index, prediction of genetic effects with mixed models, and the estimation of genetic values using REML/BLUP [20] is a potentially good strategy for selection of elite genotypes. Moreover, Silva et al. [21], Oliveira et al. [22], Rocha et al. [23] and Woyann et al. [24] opined the efficiency of this selection index technique for plant breeding. The FAI-BLUP index technique has also been utilized to assess the strengths and weaknesses of genotypes.

This study aimed to determine the genetic potential of white Guinea yam breeding genotypes using factor analysis ideotype-design (FAI-BLUP) index and multivariate techniques for selection of superior genotypes.

## 2. Results

### 2.1. Principal Component Analysis, Factor Analysis and Phenotypic Breeding Values 

The principal component analysis unravels the contributions of phenotypic breeding values of 18 metric traits measured in 36 white yam genotypes (Appendix A; Appendix A). Five principal components with eigenvalues >1.0 based on the Kaiser criterion [25] to have sufficiently explained the largest amounts of variations in the datasets (Appendix A; Appendix A). The first five PCs accounted for about 79.03% of the variations in the breeding values of the genotypes for the assessed traits. The PCA results demonstrate a trend of higher genetic values of some traits with significant strong positive or negative contributions to the observed genetic variability compared to the contributions of those with weak correlation coefficients. The breakdown value, peak viscosity and stem diameter per plant traits that associated with fresh tuber yield are to the right of PC1 (Figure 1). In the present study, the phenotypic breeding values of all traits exhibited differential contributions to variability in 36 yam genotypes (Figure 1; Appendix A). The breakdown value, peak viscosity and stem diameter per plant traits that associated with fresh tuber yield are to the right of PC1 (Figure 1).

Pairwise relationship coefficients of phenotypic estimated breeding values and genotypes are displayed in Figure 2. The phenotypic trait marker-based relationship heat-map categorized the genotypes into three main groups. The group 1 comprised of 9 members (TDr0000362, TDr04-219, TDr9902789, TDr9700917, TDr8902607, Ehobia, Nndu, TDr9519158 and TDr8902677). Group 2 comprised of 18 members (TDr9902562 TDr9501969, Leusi, TDr9518544, Pouna, Meccakusa, Pampars, TDr9700793, TDr9519177, TDr06-15, TDr9700205, Alumaco, Ojuiyawo, Yangbedu, TDr08-21-3, TDr9519156, TDr9700632 and Agbanwobe): whereas group 3 comprised of 9 members (Ufenyi, TDr9600629, TDr9902626, TDr08-21-1, TDr8902475, TDr9518988, TDr8902157 and Fakesta). The pairwise relationship coefficients vary significantly among trait-genotype associations. Genotypes TDr0000362 and TDr9519158 exhibited significantly strong positive associations for holding strength and final peak viscosity; whereas genotypes TDr9518544 and TDr08-21-3 had strong negative relationship with holding strength; and genotypes TDr9902562, TDr9501932, TDr08-21-3 and TDr9519156 exhibited strong negative relationship with final peak viscosity. Genotypes TDr08-21-2 and TDr8902665 with peak viscosity, whereas TDr9700917 had strong negative relationship with peak viscosity. Genotypes Nndu and TDr8902475 had strong negative relationship with yam mosaic virus, whereas TDr0000362 and TDr9902789 had moderate positive relationship with yam mosaic virus.

The yam ideotypes were determined through the estimation of the genetic correlation of each factor using the FAI-BLUP index. The FAI-BLUP index identified five factors. Factor 1 is associated with pasting temperature, stem number per plant, stem diameter, number of tubers per plant and ash content; whereas Factor 2 was correlated with starch property attributes (pasting time, holding strength, breakdown value and final paste viscosity); Factor 3 was associated with peel loss, starch yield, flour yield; Factor 4 was correlated with yam mosaic virus, leaf chlorophyll content and tuber yield; Factor 5 was correlated with dry matter content, peak viscosity and protein content (Table 1). The average communality and specificity accounted for 77% and 23% of all the genetic variability in the dataset, respectively (Table 1).

### 2.2. Predicted Selection Gains

Of the 18 traits studied, 15 parameters had desired genetic gains using the FAI-BLUP index (Table 2). Three traits with undesired selection gain using the FAI-BLUP index included stem diameter (−1.34%), and two starch property related traits including holding strength (−26.31%) and final paste viscosity (−3.33%). The FAI-BLUP index provided total genetic gains of 148.91% for the studied multi-traits desired for increase and –29.26% for those desired for decrease (Table 2).

### 2.3. Selection of Genotypes

Of the 36 clones evaluated, the FAI-BLUP multiple trait index identified, TDr9518544, TDr9501932, Pampars, TDr08-21-2 and TDr08-21-3 as high performing genotypes for multiple traits (Figure 3). These genotypes show the greatest potential for the simultaneous improvement of the measured traits in white Guinea yam breeding programs. Crosses involving these genotypes are expected to increase the frequency of favorable alleles in the resulting progenies while maximizing genetic variability and heterosis.

### 2.4. The Strengths and Weaknesses View of Genotypes

The strengths and weaknesses of the genotypes showed that the first factor (FA1) represented the highest contribution for genotypes Pampars and TDr08-21-2, while FA2 exhibited the highest contribution for genotype TDr9501932 (Figure 4; Table 3 and Appendix A). The FA3 had the highest contribution for genotype TDr08-21-2; and FA4 indicated the highest contribution for genotypes TDr08-21-3 and TDr9501932. 

Genotypes Pampars and TDr9518544 have strengths related to FA1 (stem number per plant, stem diameter, low yam mosaic virus, pasting temperature, ash content and number of tubers per plant). Genotype TDr9501932 is associated with the FA2 and is characterized by high flour yield alongside low disease score (Table 3). Genotype TDr08-21-2, have strengths related to FA3 (tuber yield, dry matter content and starch yield). On FA4, genotypes TDr9501932 and TDr08-21-3 displayed high traits values for peel loss and chlorophyl content. 

Genotypes with high values for positive gains and low scores for negative gains indicate their relevance in white yam population improvement for desired traits. Genotype TDr08-21-2 combined low mosaic virus infection, high fresh tuber yield, high dry matter content, high starch yield, high flour yield and food nutrient related traits. Genotypes TDr9518544 and TDr9501932 combined low peel loss, high starch yield, high flour yield, low yam mosaic virus, high leaf chlorophyll content and high fresh tuber yield. These genotypes stand out as potential genitors to obtain segregating populations with desired attribute traits.

## 3. Discussion

In this study, we assessed the genetic potential of 36 parental lines of white Guinea yam (*Dioscorea rotundata*) for multiple traits. Plant breeders often aim at gathering various desirable alleles or traits in an ideotype or one new genotype that could result in high performance and genetic gain. However, selection of ideotype genotypes based on multiple traits is often cumbersome. Multivariate techniques including principal component analysis (PCA), factor analysis (FA), and cluster analysis (CA) as well as various selection indices [26,27], the base index [28], the index of Pesek and Baker [29], and Mulamba and Mock [30] are widely being utilised for grouping of the measured traits and selection of elite genotypes, respectively. In the present study, we used a relationship heat-map clustering pattern and PCA for grouping of genotypes and traits (Figure 1 and Figure 2). Findings on the dissection of studied traits using the PCA indicated differential sets of genotypes by traits associations that could be useful in selection of parents for a breeding program. Moreover, the pattern of associations among the genetic values of the studied traits suggests that selection based on a multiple-trait selection index has potential for identifying superior breeding lines. Norman et al. [5] also found different associations between white yam genotypes, and studied traits (fresh tuber yields, tuber dry matter content and yam mosaic virus). The genotype by trait association study reveals rich information that can be used by breeders, especially in selection of parents for a breeding programme aimed at improving particular traits [31]. However, the relationship heat-map clustering pattern and PCA techniques could not identify the ideotype genotypes. For selection of genotypes based on information on multiple traits, Olivoto et al. [13] proposed a novel technique known as multi-trait FAI-BLUP index. Accordingly, when the yam genotypes were ranked based on breeding values of the studied multiple traits (Figure 3), the FAI-BLUP index selected genotypes TDr08-21-2, TDr9518544, TDr9501932, TDr8902665 and Pampars as high performing genotypes. Beside these genotypes, Ehobia was very close to the cut-off point, suggesting that this genotype could possess interesting features that could be exploited for breeding. This view is supported by the suggestion that close attention should also be paid to genotypes that are very close to the cut-off point [13]. The FAI-BLUP index model has also been utilized for the evaluation of wheat genotypes involving genetic constitutions with desirable characteristics for grain yield [32]. Our study demonstrated that the simultaneous selection for multiple traits based on the ideotype-design FAI-BLUP index contributes to the identification of high performing genotypes and expected gain from selecting those genotypes for the traits considered. This concurs with the view that the FAI-BLUP index is a potential technique for the simultaneous improvement of the multiple traits using predicted genetic effects [19].

The strengths and weaknesses view of genotypes that reveal the proportion of the FAI-BLUP index explained by each factor is an important tool utilized for identification of the strengths and weaknesses of the studied genotypes. Based on our findings, genotypes Pampars and TDr08-21-2 (FA1); TDr8902665 (FA2), genotypes TDr9501932, TDr9518544 and TDr08-21-2 (FA3) and genotypes TDr9518544 and TDr9501932 (FA4) exhibited strengths in various agronomic traits (Figure 4; Appendix A). However, genotype TDr8902665 exhibited weakness for fresh tuber yield (i.e., had the lowest value of the trait). Knowledge of contributions by these genotypes help in selection of possible putative genitors for future crossings.

## 4. Materials and Methods

### 4.1. Experimental Site

This study was conducted at the experimental site of the International Institute of Tropical Agriculture (IITA), Ibadan, Nigeria (07°29.299′′ N, 003°53.186′′ E and 224 m altitude) during the 2017/2018 cropping season. The site represents a forest savannah transition zone with total rainfall of 1305.55 mm, a mean minimum and maximum air temperature of 22.7 °C and 30.4 °C, and a mean minimum and maximum relative humidity of 54.8% and 92.4%, respectively. Soil samples of the trial site were collected and analyzed to determine the soil’s inherent fertility that supports the growth and development of the studied clones using the procedures described by ISRIC/FAO [33]. The attributes of the soil at the trial site were: slightly acidic with pH = 6.2, organic carbon (0.43%), nitrogen, N (0.005%), phosphorous, P (37.07 ppm) and potassium, K (0.758 Cmol/kg).

### 4.2. Experimental Materials, Layout, Design and Planting

Thirty-six white yam genotypes were used for the trial laid out in a 6 × 6 triple lattice design. The pedigree details of the breeding lines used for the present study are presented in Appendix A. Healthy tubers were cut into setts of 0.2 kg each, pre-treated in a mixture of 0.07 kg Mancozeb, 75 mL Chlorpyrifos and 10 L tap water for 5 min and dried for 20 h under shade. The treated setts were planted in early May 2017 in holes made on the crest of ridges spaced 1 m × 1 m between and within rows giving a population of 10,000 plants ha^−1^. Weeding was done manually using hoes.

### 4.3. Data Collection

A total of 18 agro-morphological and quality traits were collected based on protocols described by Asfaw [34] with slight modifications (Appendix A). Virus disease severity scores were collected using the 1–5 disease rating scale; where 1 = no visible symptom of the disease; 2 = mild; 3 = low; 4 = intermediate; 5 = high on 8 plants per plot at 60, 75, 90, 105, 120, 135, 165 and 180 days after planting (DAP). The disease severity score values for yam mosaic virus were converted to percentages and then used to estimate the area under disease progress curve (AUDPC) values as described by Forbes et al. [35]:(1)$$\mathrm{AUDPC}=\overset{n-1}{\underset{i=1}{\sum }}\left(\frac{y_{i}+y_{i+1}}{2}\right)\;\left(t_{i+1}-t_{i}\right)$$
where $y_{i}$ = disease severity at the *i*th observation, $t_{i}$ = time (days) at the *i*th observation, and n = total number of observations.

The dry matter content (%) was determined using the oven-dry method [36]. The starch content was measured using a modified protocol of Asaoka et al. [37]. Tuber samples of each genotype were weighed, washed, peeled, shredded and mixed in a container. About 200 mL of distilled water was added to 0.1 kg of each sample in a bottle and grinded for 5 min using the LabMill. After blending, 3 L of water was added before sieving using 125 µm mesh (Yokyo SANPO). The mixture was left for 20 h followed by decantation of supernatant. The starch particles in dishes of known masses were dried to constant mass in an oven at 60 °C.
(2)$$\mathrm{Starch}\;\mathrm{content}\;\left(\%\right)=\frac{W_{3}-W_{1}}{W_{2}-W_{1}}\times 100\%$$
where *W*_1_ = mass of evaporating dish; *W*_2_ = mass of evaporating dish + mass of starch before drying; and *W*_3_ = mass of evaporating dish + mass of starch after drying.

The pasting properties of starch were determined using a Rapid Visco-Analyzer (RVA Super 3, Newport Scientific Pty. Ltd., Warriewood, Australia). A 3 g sample of starch was dissolved in 25 mL of water in a sample canister. The sample was thoroughly mixed and fitted into the RVA as recommended [38,39].

Yam flour was produced by the method described by Komolafe and Akinoso [40]. Fresh yam tubers of 3 kg per genotype were peeled, washed, sliced, blanched in water heated at 70 °C for 15 min, oven-dried at 60 °C, milled using a hammer mill and packaged. The flour yield was determined using the protocol described by Krochmal and Kilbride [41]:(3)$$\mathrm{Flour}\;\mathrm{yield}\;\left(\%\right)=\frac{\mathrm{WDF}}{\mathrm{WFTBR}}\times 100$$
where WDF is the weight of dried flour content and WFTBR is the weight of fresh tuber.

### 4.4. Analysis

#### 4.4.1. Phenotypic Breeding Value Analyses

Each agronomic trait was analyzed using a mixed-effect model based on the following equation:(4)y=Xb+Zu+e
where y is an $n\left[=\sum_{j-1}^{r}\left(gr\right)\right]\times 1$ vector of response variable, that is the response of the *i*th genotype in the *j*th block (*i* = 1, 2, …, ց; *j* = 1, 2, …, r; $y=[y_{11},\;y_{12},\;\ldots ,\;y_{gr}]\prime )$; b a 1×r vector of unknown and unobservable block fixed effects ($b=[y_{1},\;y_{2},\;\ldots ,\;y_{r}]\prime )$; u an $m\left[=1\times g\right)$ vector of unknown and unobservable genotype random effects $\left(u=[a_{1},\;a_{2},\;\ldots ,\;a_{g}\right]\prime )$; X an n×r design matrix of 0 s and 1 s depicting y to b; Z represents an n×r design matrix of 0 s and 1 s depicting y to u; e is an n×r vector of random errors $e=[y_{11},\;y_{12},\;\ldots ,\;y_{gr}]\prime )$; and the prime symbol (’) is the vector transposition. The random vectors u and e are assumed as normal and independently distributed with a zero mean and variance-covariance matrices G and R, respectively, so that
(5)$$\left[\begin{array}{c} u\\ \varepsilon \end{array}\right]\sim N\left(\left[\begin{array}{c} 0\\ 0 \end{array}\right],\left[\begin{array}{cc} G & 0\\ 0 & R \end{array}\right]\right)$$
where the simplest forms of G and R, namely $G={\;\hat{\sigma }}_{a}^{2}I_{g}$ and $R={\;\hat{\sigma }}_{e}^{2}I_{n}$ were used, ${\;\hat{\sigma }}_{a}^{2}$ and ${\;\hat{\sigma }}_{e}^{2}$ are variances for genotypes and random errors, respectively, and $I_{g}$ and $I_{n}$ are identity matrices of order g and n, respectively. The vectors b and u were then estimated using the mixed model equation described by Henderson [42]:(6)$$\left[\begin{array}{c} ê\\ û \end{array}\right]={\left[\begin{array}{cc} X^{\prime }R^{-1}X & X^{\prime }R^{-1}Z\\ Z^{\prime }R^{-1}X & Z^{\prime }R^{-1}Z+G^{-1} \end{array}\right]}^{-}\left[\begin{array}{c} X^{\prime }R^{-1}y\\ Z^{\prime }R^{-1}y \end{array}\right]$$
where the superscript “^−1^” is the inverse of the matrices. The variance components G and R were obtained by Restricted Maximum Likelihood (REML), using the expectation-maximization algorithm [43].

#### 4.4.2. Principal Component Analysis

The principal component analysis (PCA) was computed on the trial dataset including 18 agronomic variables and 36 white yam genotypes with the aim of enhancing the discrimination power to group the measured traits based on relationships among genotypes in the studied environment. The principal component analysis (PCA) of the key agronomic traits studied was done using the stats and factoextra packages [44] in R version 3.5.3 [45].

#### 4.4.3. The FAI-BLUP Theory

The theory of the FAI-BLUP index is centered on four key steps including: (i) rescaling the traits so that all have a 0–100 range, (ii) using factor analysis to account for the correlation structure and dimensionality reduction of data, (iii) planning an ideotype based on known/desired values of traits and (iv) to compute the distance between each genotype to the planned ideotype.

##### Rescaling of Traits

The rescaling of traits was done using the following formula:(7)$$rX_{ij}=\frac{\eta_{nj}-\varphi_{nj}}{\eta_{oj}-\varphi_{oj}}\times \left(\theta_{ij}-\eta_{oj}\right)+\eta_{nj}$$
where $\eta_{nj}$ and $\varphi_{nj}$ are the new maximum and minimum values for the trait j after rescaling, respectively; $\varphi_{oj}$ and $\varphi_{oj}$ are the original maximum and minimum values for the trait j, respectively, and hij is the original value for the jth trait of the ith genotype/treatment. The values for $\eta_{nj}$ and $\varphi_{nj}$ is chosen as follows. For the traits in which negative gains are desired $\eta_{nj}=0$ and $\varphi_{nj}=100$ should be used. For the traits in which positive gains are desired then, $\eta_{nj}=100$ and $\varphi_{nj}=0$ [16]. In the rescaled two-way table ($rX_{ij}$), each column of has a 0–100 range that considers the desired sense of selection (increase or decrease) and maintains the correlation structure of the original set of variables.

##### Factor Analysis

The exploratory factor analysis was computed with $rX_{ij}$ to group correlated traits into factors and the factorial scores were estimated as follows:(8)X=µ+Lf+e
where *X* is a p × 1 vector of rescaled observations; l is a p × 1 vector of standardized means; L is a p × f matrix of factorial loadings; f is a p × 1 vector of common factors; and e is a p × 1 vector of residuals, being p and f the number of traits and common factors retained, respectively. The eigenvalues and eigenvectors are obtained from the correlation matrix of $rX_{ij}$. The initial loadings are obtained considering only factors with eigenvalues higher than one. Then, the varimax rotation criteria was used for the analytic rotation and estimation of final loadings [46,47].

The factor analysis (FA) was done to account for the dimensionality reduction of the data and relationship structure. This analysis was performed according to the following model:(9)$$F=Z{\left(A^{T}\;R^{-1}\right)}^{T}$$
where *F* is a g × f matrix with the factorial score; Z is a g × p matrix with the rescaled means; A is a p × f matrix of canonical loading, and R is a p × p correlation matrix between the traits. Furthermore, g, f, and p indicate the number of genotypes, factor retained, and measured traits, respectively. In the third step, a [1 × p] vector was considered as the ideotype matrix.

##### Ideotype

By definition based on Equation (9), the ideotype has the highest rescaled value (100) for all analyzed traits. Thus, the ideotype can be defined by a 1 × p vector I such that I = [100, 100, …, 100]. The scores for I are also calculated according to Equation (9).

##### The FAI-BLUP Index

The genetic values of traits were utilized for the estimation of the multi-trait index based on the FAI-BLUP index [19]. The desired ideotype (DI) was defined by a vector with “min” for the traits in which lower values are desired and “max” for the traits in which higher values are desired. Then the distances from each genotype according to ideotypes were estimated and converted into spatial probability as follows:(10)$$P_{ij}=\frac{\frac{1}{d_{ij}}}{\sum_{i=1;j=1}^{i=n;j=m}\frac{1}{d_{ij}}}$$
where $P_{ij}$ is the probability of the ith genotype (i = 1, 2, …, *n*) to be similar to the jth ideotype (j = 1, 2, …, *m*); $d_{ij}$ is the genotype-ideotype distance from the ith genotype to the jth ideotype based on standardized mean Euclidean distance.

The factor analysis and ideotype-design (FAI-BLUP) index were calculated for ranking of the genotypes based on multi-trait, free from multicollinearity [19]. A selection intensity of ~20 % was considered (two genotypes selected). The maximum values for tuber yield, dry matter content, leaf chlorophyll content, stem number per plant, stem diameter per plant, starch yield, peak viscosity, holding strength, breakdown value, final paste viscosity, flour yield, ash content and protein content; and minimum values for yam mosaic virus, peel loss, pasting temperature and pasting time were used for detection of ideal traits. The FAI-BLUP function of the metan package was used for the computation of the index [48]. The radar chart was generated using the radarchart function of the fmsb package [49].

##### 4.4.4. Predicted Selection Gains Analysis

The predicted genetic gain $SG\left(\%\right)$, was computed from the FAI-BLUP index for each trait considering an α% selection intensity as follows:(11)$$SG\left(\%\right)=\frac{\left({\bar{X}}_{s}-{\bar{X}}_{o}\right)h^{2}}{{\bar{X}}_{o}}\times 100$$
where ${\bar{X}}_{s}$ is the mean of the selected genotypes, ${\bar{X}}_{o}$ is the mean of the original population and $h^{2}$ is the heritability.

## 5. Conclusions

This study has provided better understanding on the genetic control and selection of the key multiple-traits in the white Guinea yam breeding program. Genotypes identified through factor analysis and ideotype-design (FAI-BLUP) index technique could be exploited as potential trait progenitors for population improvement targeted at good agronomic and end-product quality in white Guinea yam. Hybridization involving genetically diverse genotypes identified from the top 5% of superior progenies is expected to increase the frequency of favorable alleles and maximize genetic variability. The superior genotypes detected with higher genotypic values using the multi-trait selection index should be further exploited for possible commercial deployment in suitable production environments.

## Figures and Tables

**Figure 1 plants-11-03003-f001:**
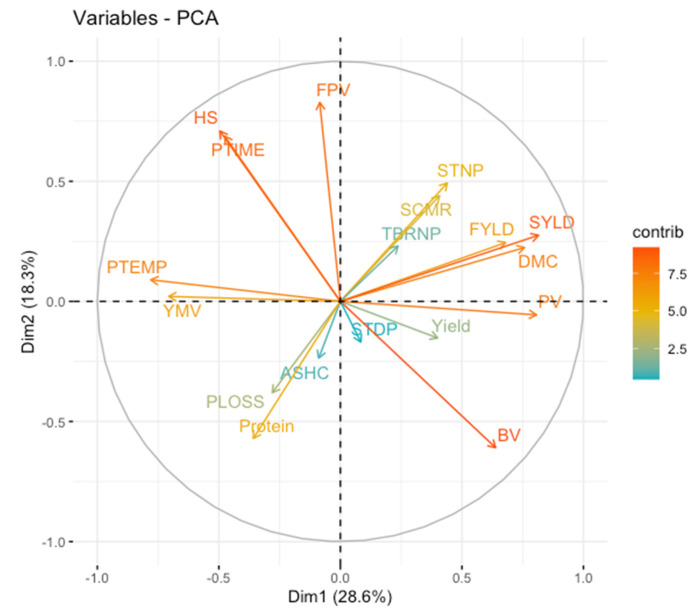
Plots showing contributions of breeding values of traits to variability in white yam genotypes. YMV: yam mosaic virus; PTEMP: pasting temperature; PTIME: pasting time; PLOSS: peel loss; SCMR: leaf chlorophyll content; STNP: stem number per plant; STDP: stem diameter per plant; TBRNP: tuber number per plant; Yield: tuber yield; DMC: dry matter content; BV: breakdown value; FPV: final paste viscosity; FYLD: flour yield; ASHC: ash content; SYLD: starch yield; PV: peak viscosity; HS: holding strength; Protein: protein content.

**Figure 2 plants-11-03003-f002:**
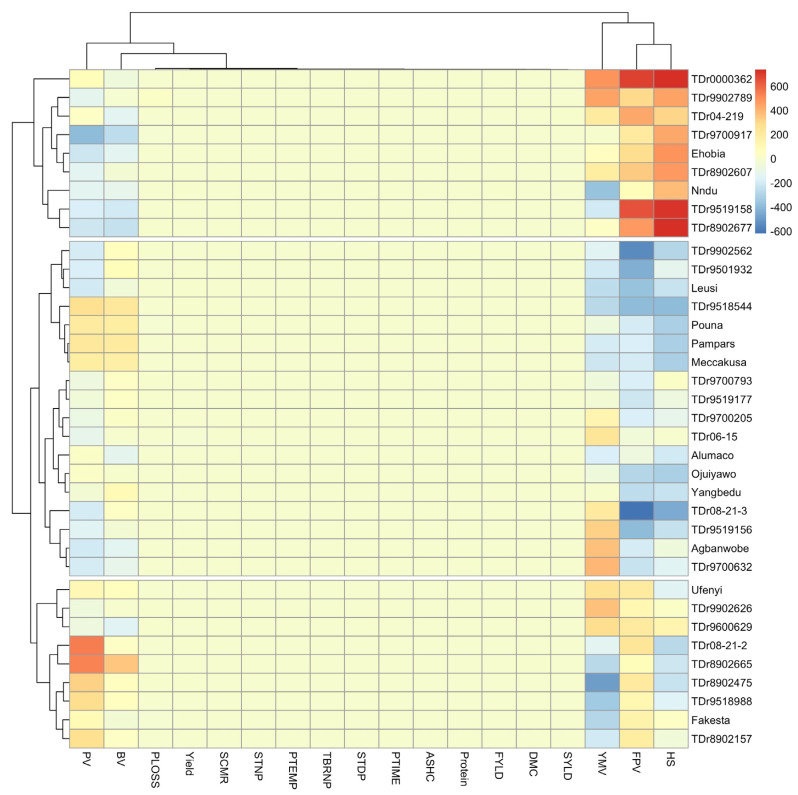
Relationship heat-map. Pairwise relationship coefficients of phenotypic estimated breeding values of 18 traits measured in 36 genotypes of yam. YMV: yam mosaic virus; PTEMP: pasting temperature; PTIME: pasting time; PLOSS: peel loss; SCMR: leaf chlorophyll content; STNP: stem number per plant; STDP: stem diameter per plant; TBRNP: tuber number per plant; Yield: tuber yield; DMC: dry matter content; BV: breakdown value; FPV: final paste viscosity; FYLD: flour yield; ASHC: ash content; SYLD: starch yield; PV: peak viscosity; HS: holding strength; Protein: protein content.

**Figure 3 plants-11-03003-f003:**
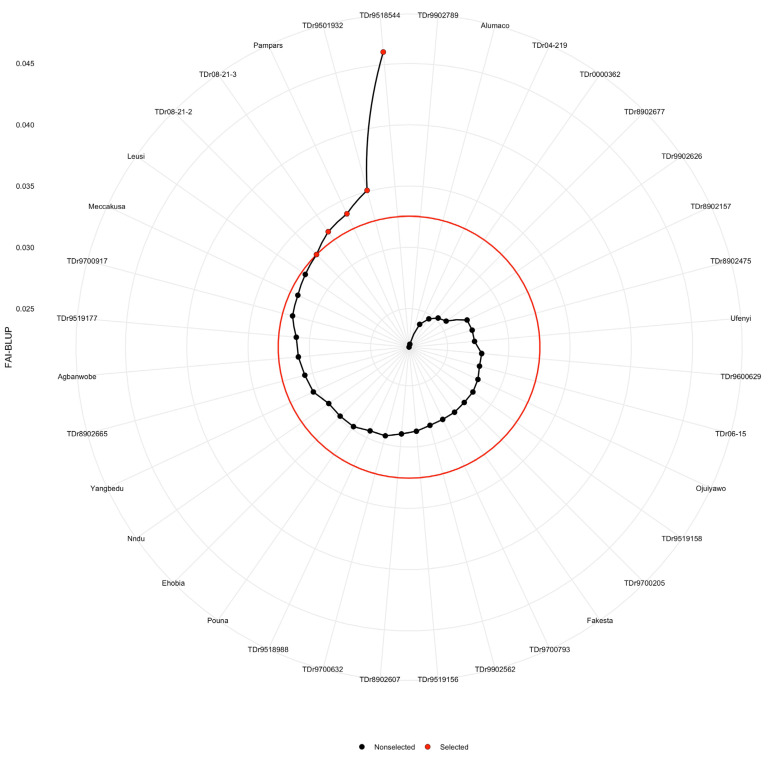
White yam genotypes ranking and genotypes selected using the factor analysis and ideotype-design (FAI-BLUP) index. The selected genotypes are shown in red and the unselected in black circles. The circle represents the cutpoint according to the selection pressure.

**Figure 4 plants-11-03003-f004:**
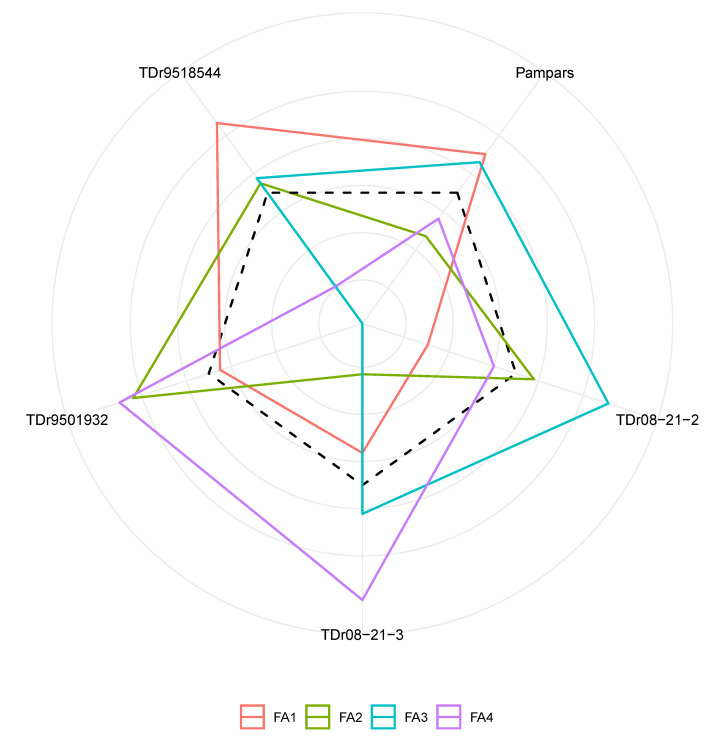
The strengths and weaknesses view of the selected genotypes is shown as the proportion of each factor on the computed factor analysis and ideotype-design distance index (FAI-BLUP). The smallest the proportion explained by a factor (closer to the external edge), the closer the traits within that factor are to the ideotype. The dashed line shows the theoretical value if all the factors had contributed equally. FA1: yam mosaic virus, stem number per plant, stem diameter, pasting temperature, ash content and number of tubers per plant; FA2: pasting time, holding strength, breakdown value, final paste viscosity; FA3: peel loss, starch yield and flour yield; FA4: yam mosaic virus, leaf chlorophyll content and tuber yield.

**Table 1 plants-11-03003-t001:** Factorial loadings, communalities, specificities and predicted genetic gains based on the factor analysis and ideotype-design (FAI-BLUP) index.

	Factors						
Trait	F1	F2	F3	F4	F5	Communality	Specificity
Yam mosaic virus	0.01	−0.21	0.34	**0.75**	−0.06	0.72	0.28
Pasting temperature (°C)	**0.52**	−0.37	0.12	0.36	−0.40	0.72	0.28
Pasting time (s)	−0.01	**−0.89**	−0.06	0.00	−0.13	0.81	0.19
Peel loss (%)	−0.04	0.19	**0.93**	−0.02	0.04	0.91	0.09
Leaf chlorophyll content	0.10	−0.06	0.01	**−0.59**	0.30	0.46	0.54
Stem number per plant	**−0.94**	0.16	−0.09	0.00	0.00	0.92	0.08
Stem diameter (mm)	**0.76**	0.18	−0.09	−0.26	−0.06	0.70	0.30
Number of tubers per plant	**−0.89**	−0.03	−0.12	0.01	0.12	0.82	0.18
Tuber yield (t ha^−1^)	−0.03	0.10	−0.01	**−0.87**	−0.12	0.78	0.22
Dry matter content (%)	−0.35	0.25	−0.25	−0.14	**0.69**	0.74	0.26
Peak viscosity (cP)	−0.40	0.49	−0.30	−0.15	0.51	0.77	0.23
Holding strength (cP)	−0.07	**−0.90**	0.04	0.13	0.04	0.83	0.17
Breakdown value (cP)	−0.24	**0.87**	−0.15	−0.09	0.14	0.86	0.14
Final paste viscosity (cP)	−0.27	**−0.77**	−0.11	0.04	0.37	0.81	0.19
Starch yield (%)	−0.19	0.23	**−0.71**	−0.03	**0.55**	0.89	0.11
Flour yield (%)	−0.01	0.13	**−0.85**	−0.30	0.16	0.86	0.14
Protein content (%)	−0.16	0.20	0.00	0.03	**−0.75**	0.64	0.36
Ash content (%)	**−0.60**	−0.08	0.28	−0.31	−0.35	0.66	0.34
Average						0.77	0.23

F1, F2, F3, F4 and F5 = factors 1, 2, 3, 4 and 5, respectively. The bold represent the traits with high (>0.5) contribution to each factor.

**Table 2 plants-11-03003-t002:** Predicted genetic gains for the factor analysis and ideotype-design (FAI-BLUP) index.

Factor	Trait	Sense	Goal	Genetic Gain (%)
FA1	Pasting temperature (°C)	Decrease	100	−0.83
FA1	Stem number per plant	Increase	100	31.99
FA1	Stem diameter (mm)	Increase	0	−1.34
FA1	Number of tubers per plant	Increase	100	16.51
FA1	Ash content (%)	Increase	100	7.03
FA2	Pasting time (s)	Decrease	100	−3.19
FA2	Holding strength (cP)	Increase	0	−26.31
FA2	Breakdown value (cP)	Increase	100	39.72
FA2	Final paste viscosity (cP)	Increase	0	−3.33
FA3	Peel loss (%)	Decrease	100	−9.53
FA3	Starch yield (%)	Increase	100	12.30
FA3	Flour yield (%)	Increase	100	6.74
FA4	Yam mosaic virus	Decrease	100	−15.70
FA4	Leaf chlorophyll content	Increase	100	0.48
FA4	Tuber yield (t ha^−1^)	Increase	100	16.43
FA5	Dry matter content (%)	Increase	100	4.71
FA5	Peak viscosity (cP)	Increase	100	9.28
FA5	Protein content (%)	Increase	100	3.73
Total (Increase)				148.91
Total (Decrease)				−29.26

**Table 3 plants-11-03003-t003:** Factorial loadings, communalities, and uniqueness of five selected genotypes based on the factor analysis and ideotype-design (FAI-BLUP) index.

	Factors					
VAR	FA1	FA2	FA3	FA4	Communality	Uniqueness
Yam mosaic virus	0.02	**0.85**	0.26	−0.46	1.00	999 × 10^−14^
Pasting temperature (°C)	**−0.55**	**−0.72**	0.33	0.28	1.00	289 × 10^−13^
Pasting time (s)	0.27	**0.96**	0.07	−0.04	1.00	444 × 10^−14^
Peel loss (%)	0.21	0.08	0.14	**−0.96**	1.00	178 × 10^−13^
Leaf chlorophyll content	−0.21	0.24	0.20	**0.92**	1.00	278 × 10^−13^
Stem number per plant	**−0.83**	−0.09	0.06	**0.54**	1.00	222 × 10^−13^
Stem diameter (mm)	**0.79**	0.43	−0.08	−0.43	1.00	200 × 10^−13^
Number of tubers per plant	**−0.83**	−0.12	−0.03	**0.54**	1.00	155 × 10^−13^
Tuber yield (t ha^−1^)	0.06	0.23	**−0.96**	−0.14	1.00	888 × 10^−13^
Dry matter content (%)	−0.15	−0.26	**0.71**	**0.64**	1.00	222 × 10^−13^
Peak viscosity (cP)	−0.33	**−0.83**	0.34	−0.28	1.00	189 × 10^−13^
Holding strength (cP)	**0.94**	0.09	−0.04	0.32	1.00	222 × 10^−13^
Breakdown value (cP)	**−0.50**	0.17	**0.85**	0.04	1.00	178 × 10^−13^
Final paste viscosity (cP)	0.24	**−0.96**	0.11	−0.12	1.00	144 × 10^−13^
Starch yield (%)	0.26	0.06	**0.96**	−0.06	1.00	0.00
Flour yield (%)	0.09	**−0.77**	**0.61**	−0.16	1.00	167 × 10^−13^
Protein content (%)	**−0.96**	0.07	−0.14	0.23	1.00	133 × 10^−13^
Ash content (%)	**−0.52**	0.41	**−0.70**	0.27	1.00	444 × 10^−14^

The bold represent the traits with high (>0.5) associated to each factor.

## Data Availability

Not applicable.

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
