# Peer review of "Multiple-Traits Selection in White Guinea Yam (Dioscorea rotundata) Genotypes"

_plants, 2022, doi:10.3390/plants11213003_

Round 1

Reviewer 1 Report

The manuscript prepared by Norman et al. evaluated the genetic/breeding potentials of 36 yam genotypes simultaneous based on 18 agronomic traits. In general, this manuscript is well-structured, and investigated question is clear. But many details are missing which made the readers cannot follow in some sections. I suggest the authors to add more details, especially in analysis method section.

1. The sentence in line 47-50 is unclear to me. In the context of linear mixed model, the BLUP is the estimation (prediction) of the random effect, and the REML is the way how the solve the mixed model to obtain the variance component. Here you just say it is an analytic procedures, which is not clear.

2. In line 82-84, you state the variance is larger in one trait comparing with others. Have you done any normalization (i.e. [0,1]) before analyzing the data? If not, the traits of large value are definitely have large variance.

3. In line 85-86, how did you define “due largely to genetic effect”? It is better to give the percentage value.

4. What is “Acgen” in Table 1? I do not find any definition of this term in the text.

5. In Figure 1 and 2, the abbreviation of train name should be given in the caption or in methods section.

6. In line 141, you just suddenly mention the “ideotype”, which the definition is unclear. In the method section, it definitely needs more description of how the FAI-BLUP works. For the general readers, the paper should be stand-alone without any further refer to original article, since this is the core methods in this manuscript.

7. In line 147, how did you correlate the factors to traits? For instance, the factor 5 is correlated to peak viscosity (r=0.51), but not with starch yield which the r=0.55. Please explain more details.

8. In Table 3, why the goal is 100, but the actually genetic gain value is negative? Thia at least should be mentioned and discussed.

9. In Figure 4, why only 4 factors? Is there something special for the factor 5?

10. In line 201-204, what is difference to the paragraph above figure 4? Do you mean this paragraph is based on the strength? Here the strength means the genetic gain?

11. In the method section, the trait data should be measured in several replicates/blocks. More details about the number of replicates for each trait should be mentioned.

12. It is better to list the sub-section in the Analysis section, i.e. each analysis method should be in one small section and ordered as they in the results section.

13. In line 337, the model notion is wrong. It should be “y=Xb+Zu+e”.  

14. In line 352, the superscript of “-” as the generalized inverse is not used on the equation. So it can be removed.

15. I could not follow the definition for the two heritabilities definition. Firstly, the broad-sense heritability should be as caption H. Secondly, what is the difference between sigma(g) and sigma(a)? More details are needed here.

Author Response

From: Prince Emmanuel Norman, PhD Plant Breeding, Sierra Leone Agricultural Research Institute (SLARI) PMB 1313, Tower Hill, Freetown, Sierra Leone

Date: 4th October, 2022

To: The Editor in Chief, Plants

Dear Sir,

RESPONSES TO REVIEWER 1 COMMENTS

On behalf of colleague authors, I write to thank the reviewers and editors of our manuscript for their great contributions that have improved our manuscript. Moreover, responses to reviewer’s comments have been addressed accordingly and are traceable in red font color.

REVIEWER 1

The manuscript prepared by Norman et al. evaluated the genetic/breeding potentials of 36 yam genotypes simultaneous based on 18 agronomic traits. In general, this manuscript is well-structured, and investigated question is clear. But many details are missing which made the readers cannot follow in some sections. I suggest the authors to add more details, especially in analysis method section.

  1. The sentence in line 47-50 is unclear to me. In the context of linear mixed model, the BLUP is the estimation (prediction) of the random effect, and the REML is the way how the solve the mixed model to obtain the variance component. Here you just say it is an analytic procedures, which is not clear.

Response: The sentence has been written with much clarity as suggested.

  1. In line 82-84, you state the variance is larger in one trait comparing with others. Have you done any normalization (i.e. [0,1]) before analyzing the data? If not, the traits of large value are definitely have large variance.

Response: Thanks so much for this keen observation Yes, normalization was done and we have mentioned how the traits were scaled in the revised manuscripts.

  1. In line 85-86, how did you define “due largely to genetic effect”? It is better to give the percentage value.

Response: 

  1. What is “Acgen” in Table 1? I do not find any definition of this term in the text.

Response: We have provided definition below the table in the revised document.

  1. In Figure 1 and 2, the abbreviation of train name should be given in the caption or in methods section.

Response: The meanings of the abbreviation of trait names have been given in the captions of Figures 1 and 2 as suggested.

  1. In line 141, you just suddenly mention the “ideotype”, which the definition is unclear. In the method section, it definitely needs more description of how the FAI-BLUP works. For the general readers, the paper should be stand-alone without any further refer to original article, since this is the core methods in this manuscript.

Response: Thanks for this observation and we have provided more details as suggested in the revised manuscript.

  1. In line 147, how did you correlate the factors to traits? For instance, the factor 5 is correlated to peak viscosity (r=0.51), but not with starch yield which the r=0.55. Please explain more details.

Response: Thanks for the observation. On the factor 5, the 0.55 was not considered because the trait was more associated with the FA3 (-0.71). We have add this one as well in the revised manuscript

  1. In Table 3, why the goal is 100, but the actually genetic gain value is negative? Thia at least should be mentioned and discussed.

Response:  Based on the breeding objective of a particular trait, a decrease or increase in trait value of a parameter is desired. For instance, for disease such as yam mosaic virus a decrease trait value is desired for selection. Thus, since the genetic gain value is negative corresponds with sense (desired breeding objective) we highlighted this in the discussion section.

  1. In Figure 4, why only 4 factors? Is there something special for the factor 5?

Response: Thanks for the observation, the figure 4 is a result of the FAI-BLUP. Based on the 5 selected clones, the number of FA was reduced to 4. We have addressed this in the revised manuscript. We have change table 3 as well. Appreciate your observation

  1. In line 201-204, what is difference to the paragraph above figure 4? Do you mean this paragraph is based on the strength? Here the strength means the genetic gain?

Response: The strengths and weaknesses view of the selected genotypes is shown as the proportion of each factor on the computed multi-trait genotype–ideotype distance index (MGIDI). The smallest the proportion explained by a factor (closer to the external edge), the closer the traits within that factor are to the ideotype. The contribution of factors for all the studied genotypes can be found in the supplementary table. Predicted Selection Gains Analysis is different from

  1. In the method section, the trait data should be measured in several replicates/blocks. More details about the number of replicates for each trait should be mentioned.

Response: Corrections have been done as suggested.

  1. It is better to list the sub-section in the Analysis section, i.e. each analysis method should be in one small section and ordered as they in the results section.

Response: The reviewer is correct and I thank him for this suggestion as well. The Analysis has been done in sections as suggested.

  1. In line 337, the model notion is wrong. It should be “y=Xb+Zu+e”.  

Response: Thanks for your inputs. The model notation has been corrected as suggested.

  1. In line 352, the superscript of “-” as the generalized inverse is not used on the equation. So it can be removed.

Response: Thanks for your inputs. The document has been corrected accordingly.

  1. I could not follow the definition for the two heritabilities definition. Firstly, the broad-sense heritability should be as caption H. Secondly, what is the difference between sigma(g) and sigma(a)? More details are needed here.

Response: Both of the heritability estimates are broad sense. However, one is based on entry mean basis and the other one mean plot basis. Both have been defined in the revised document.

To this end, I once again thank you very much for your great inputs.

Yours faithfully,

Dr. Prince E. Norman

Reviewer 2 Report

This article submitted for review (plants-1929539) presents very important genetic parameter estimations of traits in genotypes of white guinea yam (Dioscorea rotundata). Such estimation is needed for genetic improvement of crops before the appropriate genotypes  selection used as parental partners for breeding programs. This estimation allowed to identified the best genotypes of yam for selection and breeding program  due to the results obtained with FAI-BLUP index. The experiment is clearly presented and described. I suggest to accept this article for publication in Plants. 

Author Response

From: Prince Emmanuel Norman, PhD Plant Breeding, Sierra Leone Agricultural Research Institute (SLARI) PMB 1313, Tower Hill, Freetown, Sierra Leone

Date: 4th October, 2022

To: The Editor in Chief, Plants

Dear Sir,

RESPONSES TO REVIEWER 2 COMMENTS

On behalf of colleague authors, I write to thank the reviewers and editors of our manuscript for their great contributions that have improved our manuscript.

REVIEWER 2.

This article submitted for review (plants-1929539) presents very important genetic parameter estimations of traits in genotypes of white guinea yam (Dioscorea rotundata). Such estimation is needed for genetic improvement of crops before the appropriate genotypes  selection used as parental partners for breeding programs. This estimation allowed to identified the best genotypes of yam for selection and breeding program  due to the results obtained with FAI-BLUP index. The experiment is clearly presented and described. I suggest to accept this article for publication in Plants. 

Response: We thank you for the commendation.

To this end, I once again thank you very much for your great inputs.

Yours faithfully,

Dr. Prince E. Norman
